# Recasting Continual Learning as Sequence Modeling

**Soochan Lee**
Seoul National University
soochan.lee@vision.snu.ac.kr

**Jaehyeon Son**
Seoul National University
sjh9876@snu.ac.kr

**Gunhee Kim**
Seoul National University
gunhee@snu.ac.kr

## Abstract

In this work, we aim to establish a strong connection between two significant bodies of machine learning research: continual learning and sequence modeling. That is, we propose to formulate continual learning as a sequence modeling problem, allowing advanced sequence models to be utilized for continual learning. Under this formulation, the continual learning process becomes the forward pass of a sequence model. By adopting the meta-continual learning (MCL) framework, we can train the sequence model at the meta-level, on multiple continual learning episodes. As a specific example of our new formulation, we demonstrate the application of Transformers and their efficient variants as MCL methods. Our experiments on seven benchmarks, covering both classification and regression, show that sequence models can be an attractive solution for general MCL.[1]

## 1 Introduction

Continual learning (CL) refers to the ability to learn from a non-stationary stream of data. A CL data stream is typically a series of distinct tasks, and the model cannot access previous data while learning the current one. For example, in CL benchmarks based on image classification, each task typically consists of a small subset of classes. Since the model's access to the past data is restricted, standard training with stochastic gradient descent (SGD) consistently overwrites previously learned knowledge, which often leads to catastrophic forgetting. As *learning* usually implies SGD updates in the deep learning literature, most CL approaches try to combat catastrophic forgetting while relying on SGD to update model parameters. However, using SGD is generally not a necessity of CL; one may choose any update rule other than SGD.

In this regard, we point out that CL is inherently a sequence modeling problem. The basic structure of CL, which trains a model on a training stream $((x_1, y_1), \cdots, (x_T, y_T))$ and evaluates it on a test set $\{(x_n, y_n)\}_n$, can be considered as predicting the next token $y_n$ of the sequence $(x_1, y_1, \cdots, x_T, y_T, x_n)$. From this perspective, the CL process and the test-time inference are a sequence model's forward pass, with no SGD update. Instead, we can train the sequence model at the meta-level with a set of CL episodes. This meta-continual learning (MCL) framework is technically equivalent to conventional sequence model training, where a sequence model is trained on a set of sequences. Therefore, we argue that any recent and future advances in sequence models can be directly applied to MCL research. As a precaution, our approach should not be confused with continually learning a sequence model [40, 5], which is a conventional CL setting (not MCL) where each datapoint $x$ is a sequence.

As a particular instance of our new framework, we test Transformers [43], which are considered as the current state-of-the-art sequence model. In the past several years, Transformers have revolutionized various sequence modeling tasks such as language modeling [28, 29, 6], translation [43, 1], speech recognition/generation [30, 44], and protein analysis [33, 31], to name a few. When Transformer-based language models (LMs) are scaled up and trained with a massive collection of text data, they

---

[1]Code is available at https://github.com/soochan-lee/cl-as-seq

37th Conference on Neural Information Processing Systems (NeurIPS 2023).

exhibit an interesting ability called *in-context learning* [6]. Within a forward pass, they can learn new patterns and knowledge from the previous tokens in the current context and apply them to solve new tasks. Our framework maximizes this in-context learning ability of sequence models through meta-training and applies it as a solution to CL.

An immediate question may come to one's mind: if Transformer is a parallel architecture that can attend to all the training data at test time, is this even a valid CL setting? Borrowing the words of Katharopoulos et al. [17], Transformers with causal attention can be considered as recurrent neural networks (RNNs; 13). We can think of a Transformer's attention keys and values as the internal state of a recurrent model. When a new token comes in, this state is updated to include new key-value pairs. At test time, the Transformer queries the key-value pairs in the internal state, not the raw training data. Therefore, with proper implementation, using Transformers does not violate the assumptions of CL.

One drawback of standard Transformers is that the cost of handling each example grows linearly along with the internal state. Therefore, the computational complexity of handling an episode of length $T$ becomes $\mathcal{O}(T^2)$. Fortunately, there is a considerable amount of work on *efficient Transformers* [42], which have sub-quadratic complexity. Especially, the efficient Transformers with linear $\mathcal{O}(T)$ complexity maintain a constant computational cost to handle each token. In this work, we test Linear Transformer [17] and Performer [8] as such examples.

By conducting extensive experiments on seven diverse benchmarks covering both classification and regression tasks, we demonstrate that Transformers, including the efficient variants, have great potential as a general MCL approach, especially in large-data regimes. Furthermore, it is important to note that our approach is not limited to Transformers. By offering a novel perspective on MCL, we open up the possibility for other sequence models, including those that may be developed in the future, to be applied to this challenging problem.

Lastly, our approach offers a unique advantage in terms of biological plausibility. Despite its universality in modern deep learning, backpropagation has long been criticized for being biologically implausible [36, 26, 21, 12]. This is due to the lack of convincing proof to suggest that biological neurons perform the complex backpropagation process of (i) storing their activations during the forward pass, (ii) computing the error derivatives, and (iii) transmitting them backward. Our formulation is free from such criticism since the learning process is a simple forward pass of the sequence model. Only the meta-level optimization, which would have been performed by the evolutionary process in nature, is replaced by more efficient backpropagation.

## 2 Background and Related Work

### 2.1 Meta-Continual Learning (MCL)

Due to the additional layer of complexity from meta-level optimization, compounded by the absence of a standardized lexicon, there exists a high likelihood of confusion in discerning the settings of MCL and related domains. Due to the limited space, this section summarizes only the most relevant works. For a comprehensive comparison of MCL with other fields, such as continual meta-learning, we recommend referring to [39].

**Continual Learning (CL).** A continual learning episode $\mathcal{D}$ consists of a *stream* of training data $\mathcal{D}^{\text{train}} = ((x_1, y_1), \cdots, (x_T, y_T))$ and a test *set* $\mathcal{D}^{\text{test}} = \{(x_1, y_1), \cdots, (x_N, y_N)\}$ where $x_* \in \mathcal{X}$ and $y_* \in \mathcal{Y}$ are the input and target variables. The data in the training stream can only be accessed one at a time, and it is not possible to access previously seen data. If we assume a buffer that temporarily holds each task, this formulation can also cover the offline CL settings where the stream is segmented by tasks rather than examples. The training stream is generally assumed to be a concatenation of $K$ *task* streams, i.e., $\mathcal{D}^{\text{train}} = \mathcal{D}_1^{\text{train}} \oplus \cdots \oplus \mathcal{D}_K^{\text{train}}$, each of which is a stationary sequence of data sampled from a distinct distribution. The test set is the union of the task-specific test sets: $\mathcal{D}^{\text{test}} = \mathcal{D}_1^{\text{test}} \cup \cdots \cup \mathcal{D}_K^{\text{test}}$. The objective of the CL algorithm is to continually learn a good model $f_\theta : \mathcal{X} \to \mathcal{Y}$ from the stream by optimizing parameter $\theta$. For example, in the popular CL benchmarks of image classification, $f_\theta$ is a classifier that takes an image $x \in \mathcal{X}$ as input to output its class $y \in \mathcal{Y}$.

**Meta-Continual Learning (MCL).** Rather than manually crafting a CL algorithm, MCL aims to learn how to continually learn. More formally, we introduce the concept of *continual learner* $H_\eta : (\mathcal{X} \times \mathcal{Y}) \times (\mathcal{X} \to \mathcal{Y}) \to (\mathcal{X} \to \mathcal{Y})$, which is a functional that takes a data point $(x_t, y_t)$ and an old model $f_{\theta_{t-1}}$ as inputs and produces an updated model $f_{\theta_t}$. For convenience, we additionally

define another functional $G_\eta : (\mathcal{X} \times \mathcal{Y})^* \to (\mathcal{X} \to \mathcal{Y})$ that sequentially processes a training stream as input and outputs a trained model: $f_{\theta_t} = G_\eta(((x_1, y_1), \cdots, (x_t, y_t))) = H_\eta((x_t, y_t), f_{\theta_{t-1}})$. This sequential refinement of the model parameter $\theta$ is commonly referred to as the inner loop, which is the training phase of standard CL. MCL adds another layer of optimization for tuning the meta-parameter $\eta$ of the learner, which is often called the outer loop. The contents of $\eta$ can vary widely depending on the method, e.g., the initial parameters of a model, a meta-learned encoder, etc. Unlike traditional CL settings where a model is trained on a single training data stream and evaluated on a test set, an MCL setting consists of multiple CL episodes split into the meta-training set and the meta-test set, as shown in Fig. 1a. The episodes of the two meta-splits are assumed to be drawn from the same distribution. Generally, they are created by first randomly splitting available tasks into two groups, where each episode is built as a random combination of the tasks from the group corresponding to its meta-split. Since there are no overlapping tasks between the meta-splits, one cannot achieve a high meta-test score by simply memorizing all tasks in meta-training. For each episode $\mathcal{D}$, a model is produced by the learner, i.e., $f_{\theta_T} = G_\eta(\mathcal{D}^{\text{train}})$ and evaluated on the test set $\mathcal{D}^{\text{test}}$ to produce the meta-loss $\mathcal{L}(f_{\theta_T}, \mathcal{D}^{\text{test}})$. Then $\eta$ is updated by *meta*-gradient descent, i.e., SGD with $\nabla_\eta \mathcal{L}(f_{\theta_T}, \mathcal{D}^{\text{test}})$, to reduce this meta-loss during the meta-training phase. During the meta-test phase, the learner $H_\eta$ is evaluated on multiple CL episodes in the meta-test set. While this description represents the most basic form of MCL that our work is based on, meta-training can deviate from this scheme as long as the integrity of the meta-test is maintained, e.g., [16, 4].

**Prior Works in MCL.** Online aware Meta-Learning (OML; 16) splits a model into an encoder and a small prediction network on top of it. In the inner loop, only the prediction network is updated by SGD, while the encoder remains frozen. In the outer loop, the encoder and the initial parameters of the prediction network are optimized. A Neuromodulated Meta-Learning algorithm (ANML; 4) has an additional meta-learned component named neuromodulatory network with the same architecture as the encoder. Its output is passed through the sigmoid and multiplied to the encoder's output, gating some features. Meanwhile, there are several MCL approaches specialized to classification. Initially introduced as a few-shot learning method, Prototypical Network (PN; 38) has a meta-trained encoder and computes the average embedding, i.e., prototype, for each class during training. By computing the average embeddings online, PN can be directly used in MCL settings [3]. Generative MCL (GeMCL; 3) extends PN by (i) additionally modeling a diagonal covariance matrix of the embeddings for each class and (ii) adopting a Bayesian formulation.

## 2.2 Transformers

At the core of Transformers is the self-attention mechanism, which interchanges information among different positions within a sequence. In self-attention, each element in the input sequence of length $T$ is linearly projected to produce $d$-dimensional queries, keys, and values, each represented by $\boldsymbol{Q}, \boldsymbol{K}, \boldsymbol{V} \in \mathbb{R}^{T \times d}$. The attention matrix is computed as $\text{softmax}(\boldsymbol{Q}\boldsymbol{K}^{\text{T}}/\sqrt{d})$ where $\text{softmax}$ is applied row-wise. It is then multiplied to the value matrix to produce the output of the layer $\boldsymbol{O} = \text{softmax}(\boldsymbol{Q}\boldsymbol{K}^{\text{T}}/\sqrt{d})\boldsymbol{V}$. If self-attention is computed in this manner, it is more precisely called bi-directional self-attention since every token can attend to all the other tokens in both directions.

While it is used for encoding a sequence in parallel, uni-directional, or causal self-attention is generally used for decoding a sequence one token at a time. In causal attention, each token's output depends only on the preceding tokens, allowing new input tokens to be added without recomputing the existing embeddings. For efficient parallel training, this is typically implemented using a causal attention mask that zeros out the upper triangular elements in the attention matrix. Transformers that only have causal attention layers are called decoder-only Transformers [23] which have become the backbone of many large language models [28, 29, 6]. In addition to decoding, decoder-only Transformers are also suitable for encoding a streaming sequence where the tokens become available one at a time. This property makes them a perfect fit for our use case in MCL settings.

**Efficient Transformers.** One drawback of Transformers is their quadratic computational cost that arises from computing the $T \times T$ attention matrix. The computational complexity of $\mathcal{O}(T^2)$ can be a severe bottleneck in handling long sequences. Therefore, there has been extensive research on more efficient variants of Transformers. In the following, we describe the kernel-based approaches that we test in our experiments. These models have $\mathcal{O}(T)$ complexity and can be applied to decoding. Refer to [42] for a more comprehensive review of efficient Transformers.

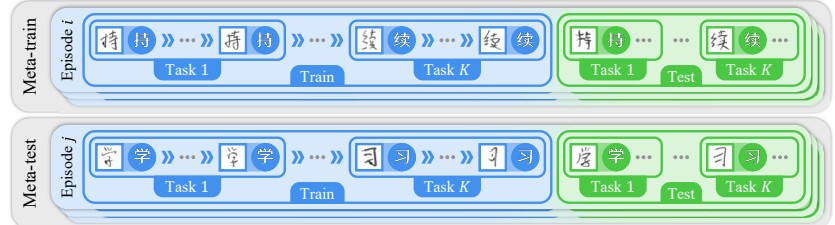

(a) The structure of an MCL dataset (examples from the CASIA benchmark)

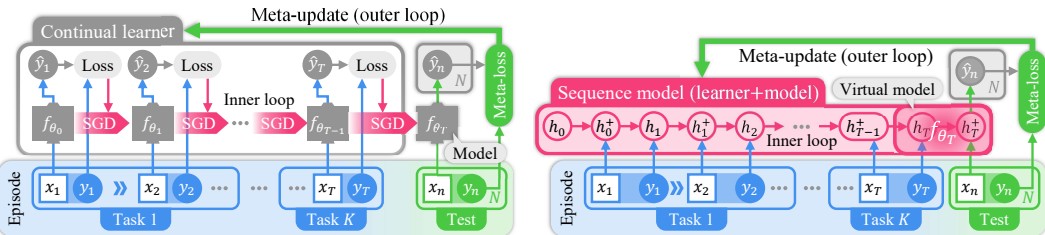

(b) SGD-based MCL algorithms      (c) CL as sequence modeling (ours)

Figure 1: Schematic illustrations of the key concepts. (a) In MCL, multiple CL episodes are split into meta-training and meta-test sets. For each CL episode, a continual learner produces a model from the training stream (blue), which is evaluated on the test set (green). The learner is meta-trained on multiple CL episodes in the meta-training set and evaluated on the meta-test set. (b) In many MCL approaches, the learner mainly depends on SGD to update the model in the inner loop. (c) In our framework, a recurrent sequence model plays both the roles of the learner and the model.

**Kernel-Based Efficient Transformers (KETs).** It approximates the attention matrix as the product of $\boldsymbol{Q}' = \phi(\boldsymbol{Q})$ and $\boldsymbol{K}' = \phi(\boldsymbol{K})$ where $\phi : \mathbb{R}^d \to \mathbb{R}^r$ is a non-linear kernel function that transforms each row of $\boldsymbol{Q}$ and $\boldsymbol{K}$ into an $r$-dimensional feature. The kernel function is meticulously designed such that $\mathrm{softmax}(\boldsymbol{Q}\boldsymbol{K}^{\mathrm{T}}) \approx \boldsymbol{D}^{-1}(\boldsymbol{Q}'\boldsymbol{K}'^{\mathrm{T}})$ where $D = \mathrm{diag}(\boldsymbol{Q}'(\boldsymbol{K}'^{\mathrm{T}}\mathbf{1}_T))$ is the normalizing term that ensures each row in the approximated attention matrix to sum to one. Through this approximation, we can change the computation order to compute $\boldsymbol{K}'^{\mathrm{T}}\boldsymbol{V}$ first: $\mathrm{softmax}(\boldsymbol{Q}\boldsymbol{K}^{\mathrm{T}})\boldsymbol{V} \approx \boldsymbol{D}^{-1}\boldsymbol{Q}'(\boldsymbol{K}'^{\mathrm{T}}\boldsymbol{V})$. The overall complexity becomes linear to the sequence length, i.e., $\mathcal{O}(T)$. Linear Transformer [17] and Performer [8] are examples of KETs and differ only in their choice of kernel function. The kernel function of Linear Transformer is defined as $\phi(\boldsymbol{x}) = \mathrm{elu}(\boldsymbol{x}) + \mathbf{1}_d$ where elu is the exponential linear unit (ELU; 9) applied element-wise. Performer [8] uses $\phi(\boldsymbol{x}) = \exp\left(\boldsymbol{W}\boldsymbol{x} - \|\boldsymbol{x}\|^2/2 \cdot \mathbf{1}_r\right)$ where the rows of $\boldsymbol{W} \in \mathbb{R}^{r \times d}$ are orthogonal random vectors.

**Decoding with KETs.** In theory, the causal attention of KET can be computed by (i) computing the approximate attention matrix $\boldsymbol{D}^{-1}\boldsymbol{Q}'\boldsymbol{K}'^{\mathrm{T}}$, (ii) applying the causal mask, and (iii) multiplying $\boldsymbol{V}$. However, this naive process requires $\mathcal{O}(T^2)$ computation, and it necessitates all tokens at once as input. Fortunately, an alternative method exists to sequentially compute the same output with $\mathcal{O}(T)$ complexity [17, 8]. Here we represent a vector corresponding to a row of the matrix as the lowercase letter, e.g., $\boldsymbol{V} = [\boldsymbol{v}_1; \cdots ; \boldsymbol{v}_T]^{\mathrm{T}}$. At each time step $t$, the new $t$-th input embedding is linearly projected to produce the query $\boldsymbol{q}_t$, key $\boldsymbol{k}_t$, and value $\boldsymbol{v}_t$. The recurrent internal state at time $t$ holds the matrix $\boldsymbol{K}_t'^{\mathrm{T}}\boldsymbol{V}_t = \sum_{t'=1}^{t} \phi(\boldsymbol{k}_{t'})\boldsymbol{v}_{t'}^{\mathrm{T}}$ and the sum of the key features $\boldsymbol{d}_t = \sum_{t'=1}^{t} \phi(\boldsymbol{k}_{t'})$. Note that both can be computed recurrently: $\boldsymbol{K}_t'^{\mathrm{T}}\boldsymbol{V}_t = \boldsymbol{K}_{t-1}'^{\mathrm{T}}\boldsymbol{V}_{t-1} + \phi(\boldsymbol{k}_i)\boldsymbol{v}_i^{\mathrm{T}}$ and $\boldsymbol{d}_t = \boldsymbol{d}_{t-1} + \phi(\boldsymbol{k}_t)$. The attention output at time $t$ is computed as $\boldsymbol{o}_t = (\boldsymbol{K}_t'^{\mathrm{T}}\boldsymbol{V}_t)^{\mathrm{T}}\phi(\boldsymbol{q}_t)/\boldsymbol{d}_t^{\mathrm{T}}\phi(\boldsymbol{q}_t)$.

## 3    Continual Learning as Sequence Modeling

Many existing MCL (and also CL) approaches rely heavily on SGD to update the model [16, 4] as summarized in Fig. 1b and Alg. 1. When each data point $(x_t, y_t)$ is made available, the model prediction $\hat{y}_t = f_{\theta_{t-1}}(x_{t-1})$ is compared with $y_t$ to produce the loss. The gradient of the loss w.r.t. the parameters $\theta_{t-1}$ is fed into an SGD-based update rule, which sometimes accompanies additional treatments to prevent forgetting, to produce updated parameters $\theta_t$.

In contrast, we propose to formulate CL as a sequence modeling problem and let a sequence model play both roles of the continual learner and the model, as presented in Fig. 1c. In our framework, the inner loop and outer loop are equivalent to the forward pass and SGD update of the sequence model, respectively. To comply with the assumptions of CL, the sequence model should be representable as a recurrent learner $H_\eta$ that iteratively refines an internal state $h_t$ as new data arrive, i.e., $h_t = H_\eta(h_{t-1}, (x_t, y_t))$. We will explain in § 3.1 that Transformers can be seen as a recurrent model. Since the Markov property holds between the internal states, $h_t$ should summarize all the data seen up to a time step $t$. The combination of $h_t$ and the sequence model's parameter $\eta$ thus defines a *virtual model* $f_{\theta_t} : (\mathcal{X} \to \mathcal{Y})$ with $\theta_t = (\eta, h_t)$. After forwarding a training stream $\mathcal{D}^{\text{train}}$ of length $T$, we obtain a virtual model $f_{\theta_T}$ trained in-context. Then, each test input $x_n$ is considered as the next token $x_{T+1}$, and $\hat{y}_n = f_{\theta_t}(x_n)$ is produced from the forward pass of the sequence model conditioned on $h_T$. The meta-loss function is defined to maximize the conditional log-likelihood:

$$\mathcal{L}(f_{\theta_T}, \mathcal{D}^{\text{test}}) = \sum_{(x_n, y_n) \in \mathcal{D}^{\text{test}}} - \log p(y_n | x_n; \theta_T) = \sum_{(x_n, y_n) \in \mathcal{D}^{\text{test}}} - \log p(y_n | x_n, h_T; \eta). \quad (1)$$

This meta-loss in the MCL perspective is technically equivalent to the popular next-word prediction loss in language modeling [28, 29]. That is, we update $\eta$ to minimize the meta-loss via SGD, as commonly done for language model training. Although we mainly assume all test examples come after the training stream, it is also possible to feed test examples in the middle of a training stream if the corresponding task is already learned. To avoid unnecessary complexity, however, we mainly assume all test examples to come after the training stream.

An important characteristic of our approach is that the target $y$ is also taken as an input of the sequence model. Since conventional MCL algorithms maintain an explicit model $f_\theta : \mathcal{X} \to \mathcal{Y}$, the only way for them to inject the knowledge from $y$ into the model is the SGD update with the inner loss $\nabla_\theta \mathcal{L}(f_{\theta_t}(x), y)$ (Alg. 1:L6). On the other hand, our sequence modeling approach replaces the SGD update with the forward pass, which incorporates $y$'s information by taking it as input. The forward pass can potentially learn a far more flexible update rule compared to SGD. As presented in Fig. 1c, we denote the state after seeing $x_t$ as $h_{t-1}^+$ to differentiate it from $h_t$, the state after seeing both $x_t$ and $y_t$.

Using sequence models can also be advantageous in terms of scalability. A major drawback of the SGD-based MCL is computing the *meta*-gradient $\nabla_\eta \mathcal{L}(f_{\theta_T}, \mathcal{D}^{\text{test}})$ in the outer loop. Its computation involves calculating higher-order derivatives and large memory for the entire computation graph of the inner loop, which holds every revision of the parameters $\{\theta_t\}_{t=1}^T$, the gradients $\{\nabla_{\theta_{t-1}} L_t\}_{t=1}^T$, and their intermediate calculations. Thus, it is hard to scale the model size or the length of the inner loop. On the other hand, our framework has no such meta-gradient (i.e., our meta-loss is a normal sequence model loss), which allows us to use a much larger and more powerful architecture within the same computational budget.

It is worth mentioning that there are other meta-learning approaches that frame various learning scenarios as the forward pass of a neural network. For instance, [35, 32] employ RNN-based architectures for few-shot learning, while [24] utilize a combination of temporal convolution and attention layers for both supervised and reinforcement learning.

## 3.1 Transformers as Meta-Continual Learning Methods

As an instantiation of our framework, we demonstrate that Transformers can be utilized as a general MCL mechanism. We only consider decoder-only Transformers, which are suitable for handling the streaming data in CL settings.

Initially, it may not be apparent how the decoder-only Transformers can be represented as a recurrent model in Fig. 1c since Transformers are frequently described as a parallel computation graph over the whole sequence. As highlighted in [17], however, Transformers with causal attention can be regarded as recurrent models. A decoder-only Transformer layer consists of two sublayers: a causal attention layer and a feed-forward layer. Since the feed-forward layers work on each token independently, information transfer across time steps occurs exclusively through the causal attention sublayer, and only in the temporal direction ($\rightarrow$). Therefore, at any time $t$, all the past information is represented as keys $\{\boldsymbol{k}_{t'}^l\}_{t'=1}^t$ and values $\{\boldsymbol{v}_{t'}^l\}_{t'=1}^t$ in each attention head $l$, which are accessed by the query $\boldsymbol{q}_t^l$. We can think of the keys and values in each attention layer as the internal state of a recurrent sequence model: $h_t = \bigcup_l h_t^l$ where $h_t^l = \{(\boldsymbol{k}_{t'}^l, \boldsymbol{v}_{t'}^l)\}_{t'=1}^t$. At every time step, this internal state is updated to

include the keys and values from the current time step, as highlighted in Alg. 2. In Alg. 2, also notice that $y_t$ is regarded as the input of Transformers (L8-9), and the virtual model $f_{(\eta, h_T)}$ is evaluated on the test set to produce the meta-loss (L11).

### 3.2 Kernel-Based Efficient Transformers for Efficient Meta-Continual Learning

Meanwhile, Transformers have been criticized for their consistently growing computational cost, which is more commonly referred to as the quadratic complexity in the Transformer literature. This limitation of Transformers triggered extensive research on *efficient Transformers* [42] which have sub-quadratic complexity. By bridging CL and sequence modeling, we allow this family of advanced Transformers to be immediately applied as CL methods. The only requirement for the model is to support decoding, which is required to sequentially take new examples in CL settings.

Among the large taxonomy of efficient Transformers, our work focuses on kernel-based efficient Transformers (KETs), which can be adapted to decoding while having $\mathcal{O}(T)$ complexity. As explained in §2.2 the decoder version of KET works very similarly to an RNN with an internal state of a constant size. As summarized in L7 and L9 of Alg. 3, the update rule corresponding to each attention head $l$ can be concisely expressed as adding the outer product of the key feature $\phi(\mathbf{k}_t^l)$ and the value vector $\mathbf{v}_t^l$ to the hidden state of a fixed size. Note that 1 is appended to $\mathbf{v}_t^l$ to compute the sum of the key features $\mathbf{d}_t^l = \sum_{t'=1}^t \phi(\mathbf{k}_{t'}^l)$ simultaneously. We test Linear Transformer [17] and Performer [8], and the only difference between the two is the kernel function $\phi$, as explained in §2.2.

**Connection to Dynamic/Static-Architecture CL Approaches.** Since the size of the internal states linearly grows as more data come in, standard Transformers can be grouped with the dynamic CL architectures. As the size and the computational cost of dynamic-architecture CL approaches typically grow linearly with the number of tasks $K$ [34, 2, 45, 20], the complexity of handling a CL episode of length $T$ is $\mathcal{O}(KT)$. In most CL scenarios where $K \propto T$, the complexity becomes $\mathcal{O}(T^2)$, which is the same as Transformers. On the other hand, KETs' internal states remain at a constant size, which makes them the counterparts of static-architecture CL approaches.

### 3.3 An Example: Continual Learning of Image Classification as Sequence Modeling

We describe a specific example of applying our approach to image classification, which has been widely used for evaluating MCL methods. In this setting, $x$ is an image,

---

**Algorithm 1** Inner loop of conventional SGD-based MCL

**Require:** learner $H_\eta$, $\mathcal{D}^{\mathrm{train}}$, $\mathcal{D}^{\mathrm{test}}$
1: $T \leftarrow |\mathcal{D}^{\mathrm{train}}|$
2: $f_{\theta_0} \leftarrow$ initial model
3: **for** $t \leftarrow 1$ to $T$ **do**
4: $\quad x_t, y_t \leftarrow \mathcal{D}^{\mathrm{train}}[t]$
5: $\quad L_t \leftarrow \mathcal{L}(f_{\theta_{t-1}}(x_t), y_t)$ $\quad \triangleright$ Loss
6: $\quad f_{\theta_t} \leftarrow H_\eta(f_{\theta_{t-1}}, \nabla_{\theta_{t-1}} L_t)$
7: **end for**
8: **return** $\mathcal{L}(f_{\theta_T}, \mathcal{D}^{\mathrm{test}})$ $\quad \triangleright$ Meta-loss

---

**Algorithm 2** Inner loop of Transformer

**Require:** Transformer $H_\eta$, $\mathcal{D}^{\mathrm{train}}$, $\mathcal{D}^{\mathrm{test}}$
1: $T \leftarrow |\mathcal{D}^{\mathrm{train}}|$
2: $\mathcal{I} \leftarrow$ indices of attention heads in $H_\eta$
3: $h_0 \leftarrow \{\}$ $\quad \triangleright$ internal state
4: **for** $t \leftarrow 1$ to $T$ **do**
5: $\quad x_t, y_t \leftarrow \mathcal{D}^{\mathrm{train}}[t]$
6: $\quad \{(\mathbf{k}_t^l, \mathbf{v}_t^l)\}_{l \in \mathcal{I}} \leftarrow H_\eta(h_{t-1}, x_t)$
7: $\quad h_{t-1}^+ \leftarrow h_{t-1} \cup \{(\mathbf{k}_t^l, \mathbf{v}_t^l)\}_{l \in \mathcal{I}}$
8: $\quad \{(\mathbf{k}_t^l, \mathbf{v}_t^l)\}_{l \in \mathcal{I}} \leftarrow H_\eta(h_{t-1}^+, y_t)$
9: $\quad h_t \leftarrow h_{t-1}^+ \cup \{(\mathbf{k}_t^l, \mathbf{v}_t^l)\}_{l \in \mathcal{I}}$
10: **end for**
11: **return** $\mathcal{L}(f_{(\eta, h_T)}, \mathcal{D}^{\mathrm{test}})$ $\triangleright$ Meta-loss

---

**Algorithm 3** Inner loop of KET

**Require:** KET $H_\eta$, $\mathcal{D}^{\mathrm{train}}$, $\mathcal{D}^{\mathrm{test}}$
1: $T \leftarrow |\mathcal{D}^{\mathrm{train}}|$
2: $\mathcal{I} \leftarrow$ indices of attention heads in $H_\eta$
$\quad \triangleright$ Superscript $l$ implies $\forall l \in \mathcal{I}$
3: $h_0^l \leftarrow \mathbf{O}_{r \times (d+1)}$ $\quad \triangleright$ internal state
4: **for** $t \leftarrow 1$ to $T$ **do**
5: $\quad x_t, y_t \leftarrow \mathcal{D}^{\mathrm{train}}[t]$
6: $\quad \{(\mathbf{k}_t^i, \mathbf{v}_t^i)\}_{l \in \mathcal{I}} \leftarrow H_\eta(h_{t-1}, x_t)$
7: $\quad h_{t-1}^{+,l} \leftarrow h_{t-1}^l + \phi(\mathbf{k}_t^l)[\mathbf{v}_t^{l\mathrm{T}}; 1]$
8: $\quad \{(\mathbf{k}_t^i, \mathbf{v}_t^i)\}_{l \in \mathcal{I}} \leftarrow H_\eta(h_{t-1}^+, y_t)$
9: $\quad h_t^l \leftarrow h_{t-1}^{+,l} + \phi(\mathbf{k}_t^l)[\mathbf{v}_t^{l\mathrm{T}}; 1]$
10: **end for**
11: **return** $\mathcal{L}(f_{(\eta, h_T)}, \mathcal{D}^{\mathrm{test}})$ $\triangleright$ Meta-loss

---

while $y$ is the corresponding class label. To feed into Transformers, both $x$ and $y$ should be encoded into a fixed-sized vector. While we can simply use a convolutional network (which can be jointly trained end-to-end) for $x$, properly encoding $y$ requires some considerations. Since a class label is essentially an abstract symbol, e.g., an arbitrary index of the class, it does not have any meaning on its own. Simply hard-assigning a trainable embedding to each class would fail, as novel classes would appear in the meta-test phase. Randomly sampling a fixed embedding every time a class appears also turns out to be inapplicable as we experiment. The random embedding is the same as an untrained token embedding, which is expected to be incompatible with the trained weights of Transformers.

The general solution used in language modeling is to express everything as a combination of known vocabulary [37]. Following the same principle, we *tokenize* a class label using a fixed vocabulary $V$ and train an embedding for each token in the vocabulary. Since class labels do not impose any constraint on how they should be tokenized, we randomly sample a unique class code $(v_1, ..., v_C)$, a random combination of the tokens $v_c \in V$, when a new class appears in an episode. Note that if the code length $C$ is greater than one, each token goes into a Transformer individually, i.e., an $(x, y)$ pair is converted to the sequence $(x_{\text{enc}}, v_1, ..., v_C)$ before going into the Transformer. At test time, the Transformer sequentially outputs the code to predict the corresponding class. To reduce complications, we keep $C = 1$ in the main experiments, and the vocabulary size is set to the maximum number of classes that can appear in an episode. Experiments with $C > 1$ can be found in Appendix B.

## 4    Experiments

We demonstrate the potential of our approach through experiments with seven benchmarks. Unless otherwise noted, each experiment is repeated five times, and the mean and standard deviation of the five runs are reported. Due to space constraints, this section contains only the essential information to understand the main result, while details, additional experiments and analyses are deferred to Appendix A, B, and C.

### 4.1    Benchmarks

To show the generality of our approach, we conduct experiments with diverse types of tasks, including both classification and regression. To construct the meta-training and meta-test set for a dataset, we first divide it into a set of tasks. The tasks are then split into two disjoint sets, one for meta-training and the other for meta-testing. The CL episodes for both meta-splits are constructed in the same manner. To build a CL episode $\mathcal{D}$, we randomly sample $K$ unique tasks. By default, we set $K = 20$, while additionally testing the $K = 100$ setting to compare performances with longer episodes. For each task $k$, the training stream $\mathcal{D}_k^{\text{train}}$ and the test set $\mathcal{D}_k^{\text{test}}$ contain five examples each (i.e., five shots). For each experiment, we meta-train for 50K steps with a batch size of 16 (i.e., 16 episodes in parallel) and meta-test with 1,024 episodes. Task identities are not provided in any case.

#### 4.1.1    Classification

For simplicity, we define each class as a task. All input images are resized to $32 \times 32$ resolution. We report classification errors in percentage as the evaluation metric.

**CIFAR-100 [18].** CIFAR-100 contains 60K images of 100 classes, with 600 images for each class. We randomly choose 60 classes for meta-training and the other 40 for the meta-test.

**Omniglot [19].** Omniglot is a handwriting dataset widely used as a benchmark for meta-learning or MCL. It has a total of 1,623 character classes gathered from 50 alphabets. The meta-train set consists of 963 classes (30 alphabets), and the meta-test set has 660 classes (20 alphabets). There are 20 images for each class, thus 19K images for meta-training and 13K images for meta-test.

**CASIA Chinese Handwriting Database (CASIA; 22).** This is another handwriting dataset that consists of 7,356 character classes in total. The classes comprise 7,185 Chinese characters, 52 English alphabets, 10 digits, and other frequently used symbols. The total number of images is 3.9M, which is 120 times larger than Omniglot. We randomly sample 1K classes for the meta-test and use the rest for meta-training. To the best of our knowledge, this dataset has not been used in meta-learning or MCL literature. However, its substantial size allows us to evaluate different aspects of MCL algorithms, which could not be examined with smaller datasets where meta-overfitting is a major factor in the evaluation. Our results suggest that the relative performance of MCL algorithms can differ in a large data regime.

**MS-Celeb-1M [10].** In addition to the CASIA dataset, we also propose to repurpose MS-Celeb-1M as a large-scale MCL benchmark. It is a facial image dataset with roughly 10M images of 100K celebrities. Similar to CASIA, we randomly select 1K classes, the celebrity IDs, for the meta-test and use the others for meta-training.

Table 1: Classification errors (%) in 20-task 5-shot MCL. Both meta-training and meta-test errors are reported to highlight the relationship between the degree of meta-overfitting and the scale of the dataset (🖼 : images, 🏷: classes).

| Type | Method | CIFAR-100 60K 🖼 / 0.1K 🏷 | | Omniglot 32K 🖼 / 1.6K 🏷 | | CASIA 3.9M 🖼 / 7.2K 🏷 | | MS-Celeb-1M 10M 🖼 / 100K 🏷 | |
|------|--------|------------|------------|-----------|-----------|-----------|-----------|------------|------------|
| | | Meta-train | Meta-test | Meta-train | Meta-test | Meta-train | Meta-test | Meta-train | Meta-test |
| Offline | Offline | N/A | $73.3^{\pm7.2}$ | N/A | $35.0^{\pm6.8}$ | N/A | $41.7^{\pm6.2}$ | N/A | $63.3^{\pm7.7}$ |
| CI | PN | $0.0^{\pm0.0}$ | $76.6^{\pm0.3}$ | $0.0^{\pm0.0}$ | $\mathbf{3.8}^{\pm0.1}$ | $0.2^{\pm0.0}$ | $\mathbf{0.4}^{\pm0.0}$ | $32.5^{\pm0.1}$ | $33.6^{\pm0.1}$ |
| | GeMCL | $0.0^{\pm0.0}$ | $76.6^{\pm0.4}$ | $0.0^{\pm0.0}$ | $\mathbf{3.8}^{\pm0.1}$ | $0.2^{\pm0.0}$ | $\mathbf{0.4}^{\pm0.0}$ | $32.1^{\pm0.1}$ | $33.3^{\pm0.2}$ |
| SGD | OML | $0.6^{\pm0.1}$ | $89.9^{\pm0.4}$ | $0.1^{\pm0.0}$ | $24.8^{\pm2.2}$ | $2.8^{\pm0.1}$ | $3.2^{\pm0.1}$ | $41.8^{\pm0.3}$ | $42.5^{\pm0.2}$ |
| | ANML | $0.4^{\pm0.1}$ | $88.1^{\pm1.4}$ | $0.0^{\pm0.0}$ | $31.0^{\pm6.2}$ | $3.7^{\pm0.5}$ | $4.3^{\pm0.5}$ | $43.8^{\pm0.3}$ | $44.8^{\pm0.4}$ |
| Ours | Transformer | $0.0^{\pm0.0}$ | $82.8^{\pm0.8}$ | $0.0^{\pm0.0}$ | $13.7^{\pm0.6}$ | $0.3^{\pm0.0}$ | $\mathbf{0.4}^{\pm0.0}$ | $29.1^{\pm0.2}$ | $\mathbf{30.0}^{\pm0.2}$ |
| | Linear TF | $0.1^{\pm0.1}$ | $83.4^{\pm0.5}$ | $0.0^{\pm0.0}$ | $36.0^{\pm1.4}$ | $0.4^{\pm0.0}$ | $0.7^{\pm0.0}$ | $31.1^{\pm0.3}$ | $\mathbf{32.4}^{\pm0.3}$ |
| | Performer | $0.0^{\pm0.0}$ | $82.9^{\pm0.3}$ | $0.1^{\pm0.1}$ | $37.1^{\pm4.6}$ | $0.5^{\pm0.0}$ | $0.7^{\pm0.0}$ | $32.5^{\pm0.5}$ | $33.7^{\pm0.2}$ |

Table 2: Classification errors (%) and regression errors of 100-task 5-shot MCL.

| Type | Method | CASIA | MS-Celeb-1M | Sine | Rotation | Completion |
|------|--------|-------|-------------|------|----------|------------|
| Offline | Offline | $70.7^{\pm3.7}$ | $87.1^{\pm3.3}$ | $0.0180^{\pm0.0038}$ | $0.699^{\pm0.049}$ | $0.1713^{\pm0.0047}$ |
| CI | PN | $\mathbf{0.8}^{\pm0.0}$ | $44.5^{\pm0.1}$ | N/A | N/A | N/A |
| | GeMCL | $\mathbf{0.8}^{\pm0.0}$ | $44.4^{\pm0.1}$ | N/A | N/A | N/A |
| SGD | OML | $6.8^{\pm0.9}$ | $54.5^{\pm0.2}$ | $0.0498^{\pm0.0004}$ | $0.524^{\pm0.087}$ | $0.1087^{\pm0.0001}$ |
| Ours | Transformer | $1.0^{\pm0.0}$ | $\mathbf{40.5}^{\pm0.1}$ | $\mathbf{0.0031}^{\pm0.0002}$ | $\mathbf{0.031}^{\pm0.001}$ | $\mathbf{0.0989}^{\pm0.0001}$ |
| | Linear TF | $2.3^{\pm0.1}$ | $45.3^{\pm0.1}$ | $\mathbf{0.0139}^{\pm0.0003}$ | $\mathbf{0.047}^{\pm0.002}$ | $\mathbf{0.1084}^{\pm0.0001}$ |

### 4.1.2 Regression

**Sine Wave Reconstruction (Sine).** Inspired by the sine wave regression problem in [16], we design a more challenging synthetic regression problem for MCL. A sine wave $\omega(\tau) = A\sin(2\pi\nu\tau + \psi)$ is characterized by amplitude $A$, frequency $\nu$, and phase $\psi$. We define the target $y$ as the values of the sine wave at 50 fixed points: $y = [\omega(\tau_1), \cdots, \omega(\tau_{50})]$. All $y$'s in each task share the same frequency and phase, while they can vary in amplitudes. We corrupt $y$ into $x$ by shifting phase and adding Gaussian noise, where the phase shift amount is sampled for each task. We report the mean squared error between $y$ and the model prediction $\hat{y}$ as the evaluation metric.

**Image Rotation Prediction (Rotation).** A model is given an image rotated by an angle $\psi \in [0, 2\pi)$ and tasked to estimate an angle $\hat{\psi}$. The evaluation metric is $1 - \cos(\hat{\psi} - \psi)$; thus, a perfect model would score 0 while random guessing would score 1.0 on average. We use the CASIA images and define each class as a task using the same meta-split.

**Image Completion (Completion).** An image completion problem involves predicting the missing parts of an image based on the available parts. Derived from our CASIA classification benchmark, we change the input $x$ to be the top half of the image and the target $y$ to be the bottom half. The evaluation metric is the mean squared error of the predicted pixel values.

### 4.2 Methods

For all methods, we use a five-layer CNN to encode image inputs and a three-layer MLP to encode vector inputs (e.g., the sine regression). All components are trained end-to-end from scratch. We group compared methods into four types: offline, class-incremental MCL, SGD-based MCL, and ours.

**Offline.** We compare the scores of offline learning since it is generally perceived as the performance upper bound of non-meta-CL. Note that MCL approaches can outperform offline learning by lever-

aging additional meta-training data. Since the model overfits to the training data we report the best test score achieved during training. We repeat each experiment ten times and report the average and standard error of mean.

**Class-Incremental MCL (CI).** We test PN [38] and GeMCL [3] as MCL approaches that are specialized for class-incremental settings. Although they cannot be applied to other settings, such as regression, they show strong performance in the classification benchmarks.

**SGD-Based MCL (SGD).** We select OML [16] as the main baseline since it can perform both classification and regression tasks. It has a two-layer MLP as the prediction network, on top of a meta-learned encoder. For 20-task 5-shot classification benchmarks, we also test ANML [4], which is an extension of OML with an additional modulatory network that regulates the features of the encoder. However, due to its extensive memory consumption, it could not be tested on longer episodes.

**Ours.** We test the vanilla Transformer [43], Linear Transformer [17], and Performer [8]. All the models share a similar architecture: 4 layers, 8 heads, and 512 hidden dimensions. Although it is tiny compared to modern language models, it is still larger than the FC layers of the baselines. Surprisingly, however, they require comparable or less GPU memory and time for meta-training in our experiments due to the lack of meta-gradient computation, as noted in §3. For more analysis on the computational cost, refer to Appendix C.

## 4.3  Results

In Table 1, we report the meta-training and meta-test errors of the 20-task 5-shot classification benchmarks to show the necessity of large-scale MCL benchmarks. In CIFAR-100 and Omniglot, all methods show a severe degree of meta-overfitting, the gap between the meta-training and meta-test scores. Meta-overfitting is more serious in CIFAR-100 where the task (class) diversity is lower. In such situations, the Transformers still perform competitively, but the baselines may achieve a better meta-test score due to less meta-overfitting. On the other hand, if more data is available, as in the CASIA and MS-Celeb-1M benchmarks, the Transformers outperform OML and ANML by a large margin and perform on par or better than PN or GeMCL, which are specialized for classification. Table 2 shows the results of longer 100-task 5-shot experiments in both classification and regression domains. In this more challenging scenario, the Transformers still perform comparably to class-incremental methods and significantly better than SGD-based approaches. The scores of offline learning mostly fall behind other MCL methods, which implies that non-meta-CL methods are not suitable for our experimental settings where the amount of data is far from abundant.

A strength of our approach, which comes with the use of Transformers, is parallel meta-training. While the baselines require a sequential SGD update on the training stream, Transformers can process the whole training stream in parallel, which can dramatically improve the meta-training efficiency once combined with massively parallel processors like GPUs. As a result, the meta-training time of Transformers is several times shorter than the baselines. The details can be found in Appendix C.

The tradeoff between the standard Transformer and the KETs is also worth discussing. In all experiments, the KETs sacrifice some level of accuracy for greater efficiency. The accuracy degradation is more severe in longer episodes, which is expected given that more information has to be fit into the internal state of the same size. This tradeoff suggests that the choice between standard and efficient Transformers ultimately depends on the required accuracy and the available computational resources.

In Appendix B, we present more experiments and analyses, including different episode configurations and model sizes. Although preliminary, the experiments with Transformers of varying sizes are particularly interesting, as the performance scales with the number of parameters. This is reminiscent of the scaling law of large language models [14], suggesting that scaling might be a potential solution to more challenging MCL problems.

## 5  Limitations and Conclusion

By formulating MCL as sequence modeling, we pave the way for incorporating advanced sequence modeling technologies into the MCL field. Given the observation that our approach demonstrates greater efficacy with increased data, it seems consistent with *The Bitter Lesson*, an argument popularized by Sutton [41]. It states that, in the long run, general-purpose learning algorithms capable of

leveraging more computation and data are likely to be more effective than human-designed algorithms. We believe our work is a step forward in this direction, generalizing the SGD-based update rules in CL with the learnable forward pass of a sequence model.

However, it is important to acknowledge that the sequence models also bring their own limitations and challenges. Efficiently handling long sequences is one such challenge. According to our experiments, even the efficient Transformers, which are designed to solve the exact challenge, have to trade off a nonnegligible amount of performance for efficiency, which implies plenty of room to improve. Nonetheless, considering the current pace of development in the sequence modeling field, we are optimistic about the potential of our approach; if a new solution is developed for sequence modeling, it can be readily integrated into our formulation.

## Acknowledgements

We thank Jaekyeom Kim, Dongjoo Kim, Dongyeon Woo, and Sangwoo Moon for their thoughtful feedback. This work was partly supported by Samsung Advanced Institute of Technology, the National Research Foundation of Korea (NRF) grant funded by the Korea government (MSIT) (No. 2023R1A2C2005573), and Institute of Information & communications Technology Planning & Evaluation (IITP) grant funded by the Korea government (MSIT) (No. 2022-0-00156, Fundamental research on continual meta-learning for quality enhancement of casual videos and their 3D metaverse transformation; No. 2021-0-01343, Artificial Intelligence Graduate School Program (Seoul National University)). Gunhee Kim is the corresponding author.

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

# A  Experimental Details

## A.1  Common Training Setups

Unless explicitly stated, all methods share the same encoder for input $x$. For image inputs, we use a five-layer CNN architecture. The kernel size is set to $3 \times 3$ in all convolutional layers, and batch normalization [15] and ReLU [25] activation are applied after each convolution. the stride is set to 2, except for the first layer which has stride 1. The channel sizes are set to 32-64-128-256-256 for the five layers. In the case of the vector input in the Sine benchmark, we use a simple three-layer MLP architecture. It has three linear layers followed by batch normalization and ReLU. For all benchmarks, we do not use any data augmentation technique.

Since there are a lot of different experimental settings in this work, we could not perform an extensive hyperparameter search in every setting. Therefore, we mainly tune the hyperparameters to the CASIA classification benchmark and apply the same settings to the other experiments. We find that Transformers are very robust to the choice of hyperparameters, achieving impressive scores in all benchmarks without any benchmark-specific hyperparameter tuning. In some baselines, however, we had to manually tune the hyperparameters specific to a benchmark, in order to achieve a reasonable score. Please refer to the configuration files and commands in the included code for the exact configurations.

## A.2  Benchmarks

**CASIA.** The CASIA dataset originally contains grayscale images of various sizes since the images are cropped tightly to the handwriting. We preprocess these images into $32 \times 32$ by (i) resizing the image such that the longer edge becomes 32 pixels long and (ii) adding paddings symmetrically to both sides of the shorter edge to fill the empty space. This maintains the original aspect ratio of the images while standardizing their size.

**MS-Celeb-1M.** MS-Celeb-1M also contains images of various sizes. Since the images are roughly aligned to the face, we simply resize them to $32 \times 32$ resolution without keeping the aspect ratio. We also exclude the classes (celebrity IDs) with fewer than 20 examples.

**Rotation.** In the Rotation benchmark, a model should produce an angle $\psi$ given an image. Due to the cyclic nature of the angle, we let the model output $(\cos \hat{\psi}, \sin \hat{\psi})$ instead of $\hat{\psi}$. To necessitate learning from each training stream, we additionally shift all $\psi$ by the same random amount sampled from range $[0, 2\pi)$. Therefore, even if there is a trivial pattern in $\mathcal{X}$ indicating orientation, e.g., a unique shape marking the top of an image, the model cannot predict the shifted $\psi$ without the knowledge from the training stream.

**Completion.** Although the output format is an image, we simply treat the output and target as a 512-dimensional vector (flattened from $16 \times 32$ image). We simply use fully connected layers for output and do not use specialized architectures, such as transposed convolution.

## A.3  Baselines

As SGD-based MCL baselines, we test OML [16] and ANML [4]. While we do our best for a fair comparison, there are several modifications made to the settings and architectures. In their original paper, OML and ANML do not maintain a full computation graph of the inner loop during meta-training to reduce the computation cost. Instead, they segment the training stream every five tasks and compute the meta-gradient only within each segment. Since they do not backpropagate through a long chain of updates, the meta-training cost is reduced at the expense of optimizing with a suboptimal objective. However, since our Transformers can backpropagate through the whole episode, comparing it with such a suboptimal configuration can be considered unfair. Therefore, in our experiments, we maintain a full computation graph of the inner loop for a fair comparison. In an effort to make the baselines more robust to the choice of hyperparameters, we also set the learning rate of the inner loop as a learnable parameter.

The model's classification output mechanism is also modified. In the original design, the weights of the output head are hard-assigned to individual classes. When learning a class, the weights corresponding to the class are randomly reinitialized, to prevent overfitting to the classes in meta-

training. Since we developed the class representation scheme in §3.3 inspired by the tokenization schemes in the language modeling domain, we use the same class representation for the baselines. Using this representation, the model can output a token in the fixed vocabulary, and when a new class is introduced, an unused token is randomly assigned to it. The model can predict a class by producing the corresponding token. Since the tokens are randomly assigned to a class, we do not need to reset the weights.

### A.4 Transformers

In the early experiments, we observed that the (meta-)training often fails to converge, especially when the episode length becomes longer. After a closer inspection of the attention pattern of the failure cases, we discovered that the attention tends to be uniformly distributed across all tokens, not transmitting any useful information to the next layer.

One possible reason is that our meta-loss (Eq. 1) is not easy to optimize. Given a test input, a well-trained Transformer should send some attention query from the test input to the keys from the training sequence of the corresponding task. However, since the meta-loss is simply a next-word prediction loss, it provides only high-level guidance without enforcing any specific attention pattern. Note that the final prediction layer and self-attention layers are two distinct mechanisms. When a Transformer is randomly initialized, it seems hard for it to notice that attending to a specific part of the training stream is useful for predicting the target. Therefore, the attention layers become "dead," which is the best thing to do if their inputs are considered random noise.

As a solution, we add an auxiliary loss called *attention loss* to explicitly guide the attention pattern in the early stage of training. For each query of a test input $x_n$ of task $k$, we maximize the log sum of its attention to the keys from all training examples of task $k$. In plain words, we guide the attention to be headed to task $k$'s training sequence, but we do not care exactly which token is attended. We apply attention loss to half of the attention heads in each layer for the first 10K steps of (meta-)training.

## B Additional Experiments and Analyses

### B.1 Experiments with ResNet Encoder

In the main text, we present the experiments with a simple 5-layer CNN encoder. This is mainly because some baselines, e.g., ANML [4], require meta-training of the whole network, including the encoder. Using more sophisticated encoders, such as ResNet [11], will increase the computational cost, which is already high due to the meta-gradient computation. Furthermore, it is hard to define the proper usage of the batch normalization [15], which is part of the ResNet architecture, in CL settings.

On the other hand, OML [16] and our Transformers do not update the encoder in the inner loop. Since the encoder is updated only in the outer loop, using ResNet with these methods does not violate the underlying assumptions of batch normalization. Here we provide the results with the ResNet-18 encoder. The ResNet-18 architecture is slightly modified from the original version since it was originally designed for handling much larger images with $224 \times 224$ resolution. We replace the first $7 \times 7$ stride-2 convolution with $3 \times 3$ stride-1 convolution, remove the next $3 \times 3$ max-pooling, and reduce the number of features in all layers by half. For the efficient Transformers, we test Linear Transformer only since it generally achieves better scores than Performer. The results are presented in Table 3. The first three columns are the classification benchmarks, while the last two are regression benchmarks.

We find that using ResNet does not make a meaningful difference in the large benchmarks using CASIA or MS-Celeb-1M. Since the images used in our experiments are small ($32 \times 32$ resolution), the simple CNN encoder seems sufficient to deliver the necessary information to the upper components. In the case of Omniglot classification, the scores of all methods even degrade due to more meta-overfitting.

### B.2 Experiments with Combinatorial Class Representation

Although we introduced the combinatorial representation for classes in §3.3, we simply keep the number of tokens for each class representation to 1 in our main experiments. In Table 4, we present the results of the 20-task CASIA benchmark with different class representations. The default setting is

Table 3: Experiments with ResNet encoder in 20-task 5-shot settings.

| Method | Omniglot | CASIA | MS-Celeb-1M | Completion |
|---|---|---|---|---|
| OML | $24.8^{\pm 2.2}$ | $3.2^{\pm 0.1}$ | $42.5^{\pm 0.2}$ | $0.1092^{\pm 0.0002}$ |
| OML (ResNet) | $27.7^{\pm 1.3}$ | $3.4^{\pm 0.1}$ | $44.1^{\pm 0.2}$ | $0.1092^{\pm 0.0002}$ |
| Transformer | $\mathbf{13.7}^{\pm 0.6}$ | $\mathbf{0.4}^{\pm 0.0}$ | $\mathbf{30.0}^{\pm 0.2}$ | $\mathbf{0.0999}^{\pm 0.0002}$ |
| Transformer (ResNet) | $25.5^{\pm 1.2}$ | $\mathbf{0.5}^{\pm 0.0}$ | $32.4^{\pm 0.2}$ | $\mathbf{0.1014}^{\pm 0.0002}$ |
| Linear TF | $36.0^{\pm 1.4}$ | $\mathbf{0.7}^{\pm 0.0}$ | $32.4^{\pm 0.3}$ | $\mathbf{0.1039}^{\pm 0.0003}$ |
| Linear TF (ResNet) | $63.6^{\pm 2.1}$ | $\mathbf{0.7}^{\pm 0.0}$ | $34.9^{\pm 0.1}$ | $\mathbf{0.1051}^{\pm 0.0001}$ |

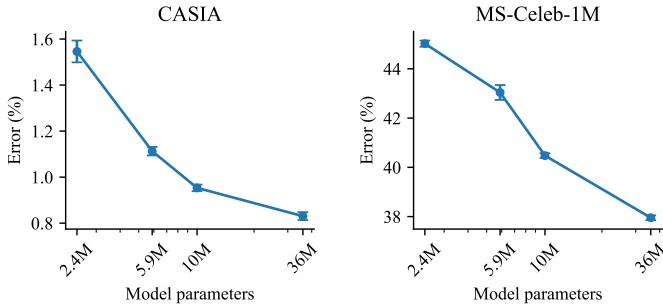

(a) Effect of scaling model size in 100-task 5-shot classification benchmarks.

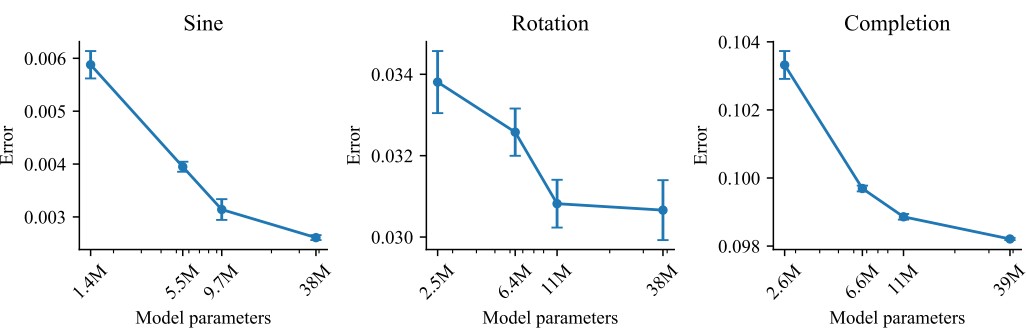

(b) Effect of scaling model size in 100-task 5-shot regression benchmarks.

Figure 2: Scaling behavior of Transformers.

represented as $|V| = 20$ and $C = 1$, where $|V|$ is the vocabulary size, and $C$ is the class representation length. We additionally test two more settings with $|V| = 8, C = 2$ and $|V| = 8, C = 3$. Note that the total number of unique class representations is $|V|^C$, which increases exponentially to $C$. We find that the classification errors are mostly similar regardless of the representation, although there is a slight performance degradation as the $C$ increases. We suspected that it is caused by the increased sequence length. Since the class representations are taken as input by the Transformer, increasing $C$ induces more computational cost. Nonetheless, we believe the idea of combinatorially representing a class or a new concept is interesting and might have a useful application in the future.

Table 4: Transformers with combinatorial class representation.

| Method | Representation | CASIA error (%) |
|---|---|---|
| Transformer | $|V| = 20, C = 1$ | $\mathbf{0.4}^{\pm 0.0}$ |
| Transformer | $|V| = 8, C = 2$ | $0.5^{\pm 0.1}$ |
| Transformer | $|V| = 8, C = 3$ | $0.6^{\pm 0.0}$ |

## B.3 Scaling Behavior of Transformers

Here we test how the model size of the Transformer is related to performance. This experiment is inspired by the scaling law of large language models [14], where the performance keeps increasing with the computational power. We test four different Transformer architectures whose number of parameters roughly ranges from 2M to 40M. The architecture details are summarized in Table 5. Note that we use the third architecture in other experiments.

In CL literature, which generally relies on SGD updates to incorporate new knowledge, increasing model size does not necessarily lead to better CL ability. On the other hand, as shown in Fig. 2a and Fig. 2b, the performance of Transformers consistently improves as more parameters are added. This is a promising result since it implies that we can simply achieve better scores by providing more computation and data. We speculate that scaling up the model size may be a necessary condition to solve more realistic and challenging scenarios.

Table 5: Transformer architecture configurations.

| Method | $n_{\text{layers}}$ | $d_{\text{model}}$ | $n_{\text{heads}}$ | $d_{\text{heads}}$ | $d_{\text{MLP}}$ |
|---|---|---|---|---|---|
| Transformer (S) | 2 | 256 | 4 | 64 | 512 |
| Transformer (M) | 2 | 512 | 8 | 64 | 1024 |
| Transformer | 4 | 512 | 8 | 64 | 1024 |
| Transformer (XL) | 4 | 1024 | 16 | 64 | 2048 |

## B.4 Forgetting Analysis

Fig. 3 and Fig. 4 show the forgetting analysis [7] of 100-task CASIA classification and Sine regression benchmarks. For each task $k$, we measure its error increase after learning $k'(> k)$ tasks, relative to the error right after learning it. Each graph in Fig. 3 and Fig. 4 shows the average of five consecutive tasks to save space. The last plot shows the average forgetting of all tasks.

Note that forgetting alone cannot be used as an evaluation metric since it measures the relative change in performance throughout an episode. For example, if a model scores zero accuracies in all tasks throughout the whole episode, the forgetting is measured as zero. Therefore, it should rather be used as an additional tool for analysis.

For the forgetting analysis, we meta-train the Transformer and Linear Transformer to also handle test examples in the middle of the training stream. For each test example of a task $k$, we randomly sample the evaluation moment from the range $[k, K]$, i.e., an example can be evaluated at the end of any task's training as long as the corresponding task is learned. In both benchmarks, Transformers' forgetting is significantly lower than OML, demonstrating the effectiveness of our approach.

## C   Computational Cost Analysis

First, we emphasize that it is hard to define a single unified measure to compare the computational cost. In some environments, the total memory consumption may be a hard constraint, while running time can be the top priority in others. Moreover, the actual cost can vary dramatically depending on the software implementation or the underlying hardware. Here we compare various aspects of the computational cost using our PyTorch [27] implementation on NVIDIA A40 GPUs which have 48 GB of VRAM. We report the cost of meta-training with a batch size of 16.

The results of the Linear Transformer and Performer are omitted due to the current limitation of our implementation. Theoretically, these efficient Transformers can be far more efficient than the standard Transformer both in terms of memory and computation time. However, realizing such advantages while maintaining high GPU utilization requires sophisticated CUDA kernel programming and cannot be implemented with simple PyTorch code. Therefore, we explicitly compute the attention matrix, the same as the standard Transformer, to save implementation time. This suboptimal implementation results in a slightly worse memory and computation time compared to the standard Transformer.

Table 6 compares the number of parameters, GPU memory consumption (in GB), meta-training speed (in episodes per second), and the classification error on the CASIA benchmark. Remarkably,

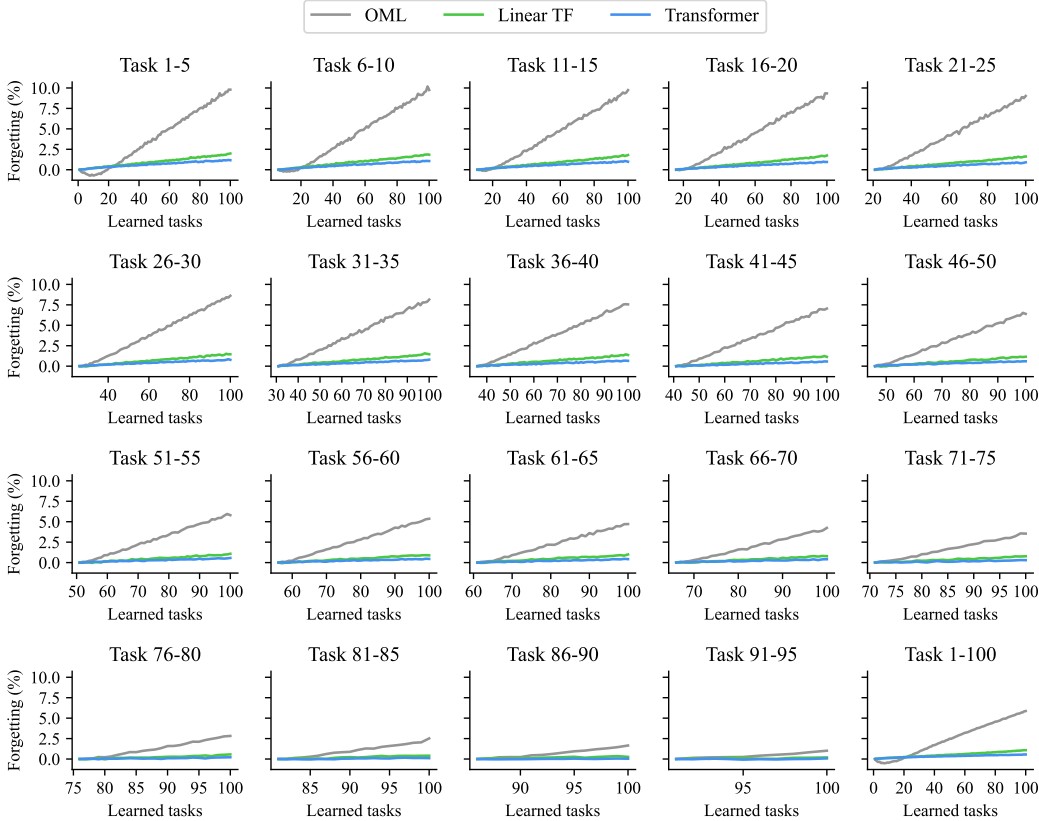

Figure 3: Forgetting analysis of 100-task CASIA benchmark.

even the largest Transformer consumes less memory than OML, the most memory-efficient baseline. Although OML has much fewer parameters than Transformers, they consume far more GPU memory due to the meta-gradient computation. If we scale up OML to have a similar number of parameters with the default Transformer architecture, which is referred to as OML (XL) in Table 6, the memory consumption increases even further, requiring four A40 with data parallelism.

Transformers are also superior in terms of (meta-)training speed. Since they can process all the examples in an episode in parallel, they can utilize GPUs more effectively. On the other hand, SGD-based baselines need to sequentially process each example and also sequentially backpropagate in the reverse direction. Therefore, Transformers have several times faster training speed compared to the baselines.

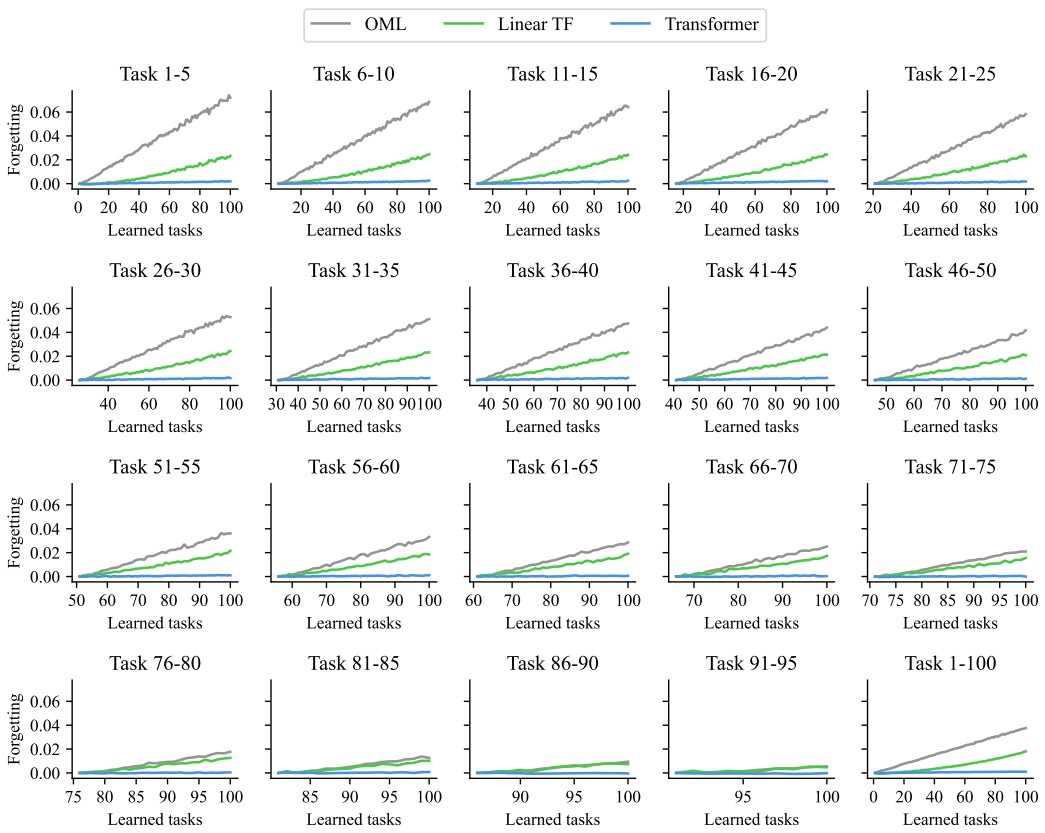

Figure 4: Forgetting analysis of 100-task Sine benchmark.

Table 6: Computational cost comparison.

| Episode | Method | Parameters | GPU memory | Training speed | CASIA error |
|---|---|---|---|---|---|
| 20-task 5-shot | OML | 1.5M | 10.7 GB | 51.2 ep/s | $3.2^{\pm0.1}$ |
| | OML (XL) | 11.5M | $4 \times 33.7$ GB | 3.5 ep/s | $1.9^{\pm0.2}$ |
| | ANML | 2.0M | 18.4 GB | 6.6 ep/s | $4.3^{\pm0.5}$ |
| | Transformer (S) | 2.4M | **4.3 GB** | **278.5 ep/s** | $\mathbf{0.6^{\pm0.0}}$ |
| | Transformer (M) | 5.9M | **4.7 GB** | **252.9 ep/s** | $\mathbf{0.5^{\pm0.0}}$ |
| | Transformer | 10.1M | **5.2 GB** | **201.5 ep/s** | $\mathbf{0.4^{\pm0.0}}$ |
| | Transformer (XL) | 36.0M | **6.2 GB** | **138.0 ep/s** | $\mathbf{0.4^{\pm0.0}}$ |
| 100-task 5-shot | OML | 1.5M | 43.7 GB | 9.40 ep/s | $6.8^{\pm0.9}$ |
| | Transformer (S) | 2.4M | **22.0 GB** | **49.69 ep/s** | $\mathbf{1.5^{\pm0.0}}$ |
| | Transformer (M) | 5.9M | **25.8 GB** | **38.78 ep/s** | $\mathbf{1.1^{\pm0.0}}$ |
| | Transformer | 10.1M | **31.5 GB** | **26.80 ep/s** | $\mathbf{1.0^{\pm0.0}}$ |
| | Transformer (XL) | 36.0M | **41.4 GB** | **16.34 ep/s** | $\mathbf{0.8^{\pm0.0}}$ |

