# OpenReview forum: "Recasting Continual Learning as Sequence Modeling"
_NeurIPS.cc/2023/Conference — NeurIPS 2023 poster_

### Official Review · Reviewer_JJ3m · 2023-07-03

**Soundness:** 3 good
**Presentation:** 3 good
**Contribution:** 3 good
**Rating:** 6
**Confidence:** 5

**Summary:**

The paper proposes to formulate continual learning as a sequence modeling problem. This new formulation views the model's learning process with continual session data as the forward pass of a sequence model, such as a Transformer, rather than relying on backpropagation. Specifically, keys and values within the Transformer can be interpreted as the internal state of a memory module in continual learning. To optimize the sequence model's parameters for continual learning problems, meta-continual learning is employed. This involves meta-learning the Transformer model using episodic learning. To address the computational and memory costs associated with long sequence modeling, efficient transformers, including the Linear Transformer and Performer, are utilized. The proposed approach's effectiveness is substantiated through performance evaluations across multiple benchmark datasets.

**Strengths:**

* Formulating continual learning as the problem of sequence modeling is very novel to me, and the author provides a very detailed explanation of why that two problems can be connected with each other under the framework of meta-continual learning. For example, the similarities between the inner loops of the two problem formulations are illustrated clearly in Algorithm 1 and 2. Furthermore, the paper conceptually outlines the connection between dynamic-architecture CL approaches and standard Transformers.

* The introduction is written in a very clear manner, making it easy for readers to grasp the central ideas and the contributions of the study.

**Weaknesses:**

* The paper doesn't adequately distinguish between continual learning and meta-continual learning. While the text mentions that meta-continual learning aims to automatically learn how to continue learning as opposed to standard continual learning by manual design, it fails to mention the necessity of a large-scale offline dataset to create a meta-training dataset. In contrast, standard continual learning does not require such an offline dataset for training.

* The explanation of meta-continual learning lacks clarity. I recommend that the authors define and explain terms such as episodes, meta-train, and meta-test more explicitly. Particularly, the concepts of a support set and query set, which are typically used under the meta-learning framework, are not mentioned in this paper.

* An important baseline in meta-continual learning, "Wandering within a world: Online contextualized few-shot learning," presented at ICLR 2021, is overlooked in the related work section.

* While the text acknowledges the limitations of SGD-based meta-continual learning, such as the high cost of second-order gradients and scalability issues, it doesn't reference previous work on efficient meta-learning approaches. e.g. [1].  I would suggest the authors incorporate references to efficient meta-learning techniques and add a dedicated section comparing them to transformers in terms of computational and memory costs.

    [1] Large-Scale Meta-Learning with Continual Trajectory Shifting. ICML 2021


* The presentation of the experiment results could be improved. A central challenge of continual learning is the balance between catastrophic forgetting and rapid adaptation to new knowledge. However, Table 1 doesn't illustrate the proposed method's effectiveness related to these two criteria. Therefore, I suggest the authors represent the experiment results using a plot charting the number of continual sessions against average accuracy. Such a plot is typically employed in previous works.

**Questions:**

Please see questions and suggestions in Weaknesses.

**Limitations:**

The paper include the limitations of the paper but fails to mention potential negative social impacts.

---

> ### Author Rebuttal · Authors · 2023-08-09
>
> We thank the reviewer for the detailed and insightful comments.
>
> #### **Continual Learning (CL) vs. Meta-Continual Learning (MCL)**
> As pointed out by the review, a large meta-training dataset is one of the fundamental assumptions of meta-learning and its variants, such as meta-continual learning. We will make sure to state this assumption clearly in the final draft.
>
> #### **Explanation of MCL**
> We will improve the description of MCL and terminologies following the suggestions. We did not use the terms “support set” and “query set” because (i) the concept corresponding to the support set in MCL is not a set but a stream, and (ii) prior works in MCL (e.g., Javed et al., Beaulieu et al.) did not use them either. But, it is a good idea to draw a connection to the meta-learning terminologies for better understanding. We will update the description accordingly.
>
> #### **Wandering Within a World (Ren et al.)**
> We thank the reviewer for introducing an important related work. However, we find this work is closer to continual meta-learning (CML) rather than meta-continual learning (MCL). These two learning frameworks sound very similar, but they are fundamentally different in terms of their underlying assumptions and objectives. Given the significant confusion they have engendered within the research community, we put a considerable amount of effort into Appendix A, where we summarize and compare various learning frameworks using visual illustrations and formal algorithms.
>
> Borrowing the expression from Ren et al., the MCL setting is *episodic*. There are multiple episodes in the meta-training set, and each episode consists of a series of tasks. For each episode, an independent model is produced, which does not need to perform well on tasks from other episodes. In CML, on the other hand, there is no meta-training / meta-test distinction, and there is only one episode. Please refer to Appendix A for more detailed illustrations. We will include Ren et al. in the prior works on CML.
>
> #### **Continual Trajectory Shifting (Shin et al.)**
> This is also an important related work. There has been little work on applying efficient meta-learning techniques to the MCL domain, and we believe this direction would be an interesting research topic in the near future. In Appendix C.6, we discussed the first-order approximation in the MCL setting. We will expand this section to more comprehensively cover various efficient meta-learning methodologies such as Shin et al.
>
> #### **Forgetting Analysis**
> In Appendix C.7, more specifically in Figures C.10 and C.11, we presented a detailed analysis of forgetting. Please understand that the sizes of the plots are too large to be in the main text, considering the page limit.
>
> #### **Potential Negative Social Impacts**
> Since our work is not directly related to applications that are immediately deployable in society, we believed that the potential negative societal impacts would be minimal. However, we will address potential negative impacts in the final version.

---

> > ### Comment · Reviewer_JJ3m · 2023-08-18
> >
> > Thanks for the author's response. Since most of my concerns are addressed, I would recommend an accept and maintain my original score.

---

> > > ### Author Response · Authors · 2023-08-20
> > >
> > > We greatly appreciate the reviewer’s recommendation for acceptance. We eagerly anticipate presenting our work at NeurIPS.

---

### Official Review · Reviewer_3kpM · 2023-07-04

**Soundness:** 3 good
**Presentation:** 3 good
**Contribution:** 3 good
**Rating:** 6
**Confidence:** 3

**Summary:**

The paper redefines Meta-Continual Learning (MCL) as a sequence learning problem. Following this definition, the paper proposes a Transformer-based meta continual learner. The method is evaluated on several classification and regression tasks.

**Strengths:**

Overall, I think the paper is well written and does a good job justifying the approach.
The idea of casting MCL as a sequence learning problem is novel (at least in CL) and interesting.

**Weaknesses:**

baselines: the paper mentions that Prototypical Networks could be used in MCL. It would be interesting to compare against them.
In general, the baselines are a bit limited. It would be interesting to compare against other methods that keep all the data in memory, like Transformers do.

minor comments:
- line 40-41 claims that sequence learning exploits in-context learning (ICL). However, ICL is a property of large pretrained models, I don’t think it applies to this setting


**Questions:**

- LINE 270: use pretrained model. Is this model trained from scratch on each episode?
- LINE 303: “task identities are not provided”. However, only the data from a specific task is provided, so it’s equivalent to having task identities.
- how robust is this algorithm to the data order?
- it is unclear to me what is the computational cost of the method at inference time compared to static networks.

**Limitations:**

-

---

> ### Author Rebuttal · Authors · 2023-08-09
>
> We thank the reviewer for the thoughtful and insightful comments.
>
> #### **Baselines**
> Since Prototypical Network (PN) and GeMCL cannot be applied to domains other than classification, we prioritized testing other baselines that can perform both regression and classification in our initial submission. As suggested, we have tested the Prototypical Network (PN) and also GeMCL (it is a relatively simple extension of PN) on the classification benchmarks and found an interesting trend. We will include the results in the final version. In the following, we compare the classification errors of PN and GeMCL in the 20-task classification benchmarks.
>
> | Method | CIFAR-100 | | CASIA | | MS-Celeb-1M | |
> |---|---:|---:|---:|---:|---:|---:|
> | | Meta-train | Meta-test | Meta-train | Meta-test | Meta-train | Meta-test |
> | PN | $0.0^{\pm0.0}$ | $76.6^{\pm0.3}$ | $0.2^{\pm0.0}$ | $0.4^{\pm0.0}$ | $32.5^{\pm0.1}$ | $33.6^{\pm0.1}$ |
> | GeMCL | $0.0^{\pm0.0}$ | $76.6^{\pm0.4}$ | $0.2^{\pm0.0}$ | $0.4^{\pm0.0}$ | $32.1^{\pm0.1}$ | $33.3^{\pm0.2}$ |
> | OML | $0.6^{\pm0.1}$ | $89.9^{\pm0.4}$ | $2.8^{\pm0.1}$ | $3.2^{\pm0.1}$ | $41.8^{\pm0.3}$ | $42.5^{\pm0.2}$ |
> | ANML | $0.4^{\pm0.1}$ | $88.1^{\pm1.4}$ | $3.7^{\pm0.5}$ | $4.3^{\pm0.5}$ | $43.8^{\pm0.3}$ | $44.8^{\pm0.4}$ |
> | MetaFSCIL | $34.5^{\pm2.1}$ | $82.1^{\pm0.3}$ | $12.0^{\pm0.4}$ | $12.2^{\pm0.5}$ | $57.6^{\pm0.3}$ | $57.8^{\pm0.2}$ |
> | Transformer | $0.0^{\pm0.0}$ | $82.8^{\pm0.8}$ | $0.3^{\pm0.0}$ | $0.4^{\pm0.0}$ | $29.1^{\pm0.2}$ | $30.0^{\pm0.2}$ |
> | Linear TF | $0.1^{\pm0.1}$ | $83.4^{\pm0.5}$ | $0.4^{\pm0.0}$ | $0.7^{\pm0.0}$ | $31.1^{\pm0.3}$ | $32.4^{\pm0.3}$ |
> | Performer | $0.0^{\pm0.0}$ | $82.9^{\pm0.3}$ | $0.5^{\pm0.0}$ | $0.7^{\pm0.0}$ | $32.5^{\pm0.5}$ | $33.7^{\pm0.2}$ |
>
> Due to their simplicity, PN and GeMCL are robust to meta-overfitting and significantly outperform all other methods in smaller datasets such as CIFAR-100. However, in the CASIA benchmark, where a larger number of classes reduces the effect of meta-overfitting, their performance is on par with Transformers. Finally, in the most challenging MS-Celeb-1M benchmark, they fall behind Transformers. We suspect that PN's simple algorithm of averaging the embeddings is not sufficient to integrate the information of the training stream if the task distribution becomes more complex.
>
> #### **In-Context Learning (ICL)**
> ICL is often introduced as an *emergent* ability of pretrained LLMs. However, considering the original description in the GPT-3 paper, the definition of ICL does not need to be restricted to pretrained LLMs. One may explicitly train an arbitrary sequence model to perform ICL, which is exactly what we do in this work.
>
> ---
> ### Questions
>
> #### **Line 270**
> Thank you for pointing this out. There is absolutely no SGD update in an inner loop. The parameters of both CNN and Transformer are fixed inside each episode, and only the outer loop updates the parameters. We will make this clear in an updated version.
>
> #### **Line 303**
> We would like to clarify our use of terminology. In the statement 'task identities not being provided,' we referred to the distinction between task-aware and task-agnostic CL. Similar to prior works on MCL, such as OML and ANML, we followed task-agnostic CL settings.
>
> Note that the term "task" has different meanings in meta-learning and MCL literature. In the meta-learning literature, a meta-training (or meta-test) set is often said to hold multiple *tasks*, but we use the term “episode” to refer to the corresponding concept in MCL. In MCL, each episode is a CL problem, which consists of $K$ *tasks* randomly sampled from the set of possible tasks. During the test phase of an episode, the model receives an input and is tasked to infer the corresponding output without knowing which task it belongs to. Since the input can belong to any of the $K$ tasks, it is not equivalent to having task IDs. We will improve the description in an updated version.
>
> #### **Robustness to Data Order**
> This is an intriguing question. Unlike conventional (meta-)continual learning methods, we can rephrase the question of “how robust the MCL method is to the data order” as “how robust the sequence model’s ICL capability is to the order of in-context examples.” Thus, the robustness depends on which sequence model is used and how the meta-training set (i.e., a training set from the perspective of sequence modeling) is constructed. We believe as long as Transformer is used as the sequence model and the meta-training set sufficiently covers diverse data orders, our approach should be robust to the data order.
>
> #### **Computational cost**
> Since our main idea is to use generic sequence models as MCL methods, the computational cost depends on the choice of the sequence model. In this work, we tested the standard Transformer, Linear Transformer, and Performer, whose computational costs per test example are $O(T)$, $O(1)$, and $O(1)$, respectively, where $T$ is the number of training examples in an episode. Note that these are the same as the costs of the sequence models inferring one token, given a context of length $T$.

---

### Official Review · Reviewer_FrpD · 2023-07-06

**Soundness:** 3 good
**Presentation:** 4 excellent
**Contribution:** 2 fair
**Rating:** 4
**Confidence:** 4

**Summary:**

This paper applies transformers and their efficient variants as sequence models to the problem of meta-continual learning. More specifically, instead of running gradient descent on a stream of training data, this paper trains transformers to do in-context continual learning over the data stream. It then compares these transformer-based approaches to three other MCL baselines on regression and image classification tasks.

**Strengths:**

Originality: There have been many works on using sequence models for meta learning, and many works on meta-continual learning. However, to my knowledge this is the first paper where sequence model based meta-learning is tested for continual learning capability.
Quality: code included, good reproducibility; rich content in the Appendix, many details and analysis.
Clarity: well-written and easy to follow. Many schematic illustrations that make the core ideas clear.
Significance: in the long run, being able to leverage data and compute to learn a CL algorithm instead of human-engineering one is an important topic.

**Weaknesses:**

The main weakness is that the experiments have not convincingly demonstrated the proposed method is practically useful as a continual learning method.

One reason is its scalability to longer sequences. If I understand correctly, during the meta-test, almost all methods are evaluated on episodes of only 20 x 5 = 100 examples. The only experiment that’s slightly longer is Table 2 with 100 x 5 = 500 examples, and only one baseline was included. In comparison, OML showed it can scale to 1000 examples and ANML to 9000 steps. But frankly even these are too short to be useful for continual learning, which is supposed to handle much longer streams than normal deep learning settings. How would this method fare if at test time the episodes are much longer?

The second reason is its generalization ability to out-of-distribution data during meta-test. All evaluations in this work assume the same distribution and the same episode length for meta-train and meta-test. However, in real CL applications, one can’t know beforehand the distribution of future data and how long the stream would be. One advantage of SGD-based continual-meta learning approach is that the inner loop uses SGD for optimization, so even if the meta-test data is OOD, SGD can still learn. Can transformers still learn new data if they are OOD?

The third reason is that it’s not compared to any competitive conventional CL baselines. Even if the authors only intend to show competitiveness among MCL approaches, it’s still good to have other types of CL approaches for reference. In addition, there are other meta-learning methods that could be competitive baselines but not referenced, for example [1, 2]. Although these meta-learning methods are not particularly designed for continual learning, they have both demonstrated continual learning capabilities. In particular, [1] proposed an optimization-based approach that can extend to arbitrarily long inner-loops and trained a continual learning optimizer with it; [2] notably also applies a transformer as a meta learner over episodes for in-context RL.

[1] Meta-Learning with Warped Gradient Descent https://arxiv.org/pdf/1909.00025.pdf
[2] In-context Reinforcement Learning with Algorithm Distillation https://openreview.net/forum?id=hy0a5MMPUv

**Questions:**

1. Why is the 100-task MCL only tested on CASIA?
2. The authors mentioned that for OML and ANML, full backprop is used instead of truncated backprop for fair comparison. But truncated backprop was what made these methods scalable to long inner loops, so for the 100-task MCL I think it’s okay to use the truncated version.
3. One advantage of OML that was highlighted in the original paper is that it’s complementary to other CL methods. Can transformer-based MCL methods be combined with other CL methods too?


**Limitations:**

As mentioned in the weakness, the main limitation of this method is that it can’t be used for continual learning in practice. I think this paper can be dramatically improved if the authors can run experiments where meta-test episodes are very long and use different distribution and episode length for meta-train and meta-test. It would also be more convincing to compare with more competitive CL baselines.

---

> ### Author Rebuttal · Authors · 2023-08-09
>
> We thank the reviewer for the thoughtful review and the acknowledgment of the various strengths of our work. We believe there are some differences in viewpoints between ourselves and the reviewer concerning the relationship between CL and MCL settings. We hope our following responses help close the gap.
>
> #### **Continual Learning vs. Meta-Continual Learning**
> We think the relationship between CL and MCL directly corresponds to the relationship between standard learning and meta-learning. Therefore, CL and MCL are two distinct learning frameworks, each with its own set of assumptions. As meta-learning methods cannot be evaluated in the standard learning setting (there is no other episode to meta-train on), MCL methods should also be compared in the MCL settings as well. Therefore, we present our method as an effective MCL approach rather than a CL method, as prior works on MCL (e.g., OML, ANML) did.
>
> #### **Scalability of Learning Paradigms Based on Meta-Learning**
> The reviewer’s concern about the scalability of MCL methods is a valid point, and we agree that scaling up MCL approaches is an important future work. However, we think the scalability issue does not stem from individual methods but the general meta-learning setting (not limited to MCL). The two-level optimization (inner loop + outer loop) of meta-learning is inevitably resource-intensive, and the size of the inner loop has to be kept small to fit the whole optimization process within an available computational budget.  Therefore, existing meta-learning (and MCL) research mostly assumes rather small-scale episodes.
>
> Given this perspective, more advances in hardware may be a prerequisite to solving the scalability issue of meta-learning, just like the remarkable progress of large language models (LLMs) enabled by the development of more powerful GPUs and distributed training system. It is important to note that none of the MCL methods (including the baselines) tested in this work has a theoretical limit on the size of the problem it can handle. Given a more computational budget, all the methods can be applied to bigger problems.
>
> In addition, we believe our sequence modeling approach has better potential in terms of scalability compared to SGD-based approaches for the following reasons. First, efficiently handling longer sequences is one of the top-priority topics in sequence modeling research, often motivated by the potential applications to LLMs. Our approach can directly benefit from the advances in this field of research. Second, both Transformers and efficient Transformers can compute the inner loop in parallel, unlike the SGD-based approaches that require sequential computation of the inner loop. Therefore, it is easier to take advantage of new hardware with massive parallelism.
>
> #### **Handling OOD Data in Meta-Test**
> Although it is desirable to generalize OOD data in the meta-test phase, it is not the primary goal of either meta-learning or meta-continual learning. The episodes in both meta-training and meta-test sets are generally assumed to be drawn from the same distribution, just like the examples in both training and test sets are assumed to be drawn from the same distribution in standard learning settings. This assumption is also stated in widely circulated works on meta-learning, such as MAML (Finn et al.) or "Meta-Learning in Neural Networks: A Survey" by Hospedales et al.
>
> It is also debatable whether SGD-based MCL approaches are truly capable of handling OOD data. As pointed out by the review, SGD update will surely enable learning new data even if it is OOD. Our main concern is whether the SGD-based approaches can effectively prevent the forgetting of such OOD data. In the case of OML, for example, it meta-trains an encoder to produce special features that are robust to forgetting from SGD. However, there is no guarantee that the encoder is still effective for OOD data.
>
> #### **Comparison with Conventional CL Methods**
> Not only in our work but also in OML and ANML papers, CL baselines are NOT compared because CL methods cannot achieve meaningful scores in MCL settings. As we explained above about the scalability issue, the current MCL settings generally focus on few-shot settings, which are unsuitable for general CL algorithms. Meanwhile, we will thoroughly discuss [1] and [2] as the review suggested.
>
> ---
> ### Questions
> #### **100-Task Experiments**
> We provided more 100-task experiments in Appendix C.2. Please understand that we had to select only a few key results for the main text due to the page limit. Appendix C also contains many other experiments.
>
> #### **Truncated Backprop for OML and ANML**
> Yes, one may use truncated backprop to improve the scalability of OML or ANML. However, as truncated backprop is an approximation of the full backprop, it can cause larger errors in OML and ANML. Thus, we chose to use full backprop in all methods.
>
> #### **Combination with Other CL Methods**
> Although it depends on the specific formulation of individual CL methods, the model updates of our approach are based on the forward pass of a sequence model (instead of SGD) and thus would be incompatible with SGD-based CL methods.

---

> > ### Comment · Reviewer_FrpD · 2023-08-19
> > **CL vs. MCL**
> >
> > My sincere apologies for the delay in my response!
> >
> >
> > I would like to express my gratitude for the comprehensive rebuttal provided by the authors, and I do think applying sequence models such as transformers to meta-continual learning is an important direction. However, it is evident that a fundamental difference exists between our viewpoints. And after a thorough examination of the authors' rebuttals, I believe that the gap still remains.
> >
> >
> > The authors assert that CL and MCL are two distinct frameworks and should not be directly compared. They emphasize that their work primarily focuses on presenting an effective MCL method, rather than a CL method. However, I hold the perspective that MCL is merely one approach to CL, achieved by learning to continually learn using some meta-training data. Its ultimate objective is no different than other CL approaches, such as those based on regularization, rehearsal, and expansion. Thus, the effectiveness of an MCL method hinges on its capability as a CL method.
> >
> >
> > The authors tried to draw an analogy between MCL and Meta-learning, contending that MCL should be permitted to make additional assumptions, such as the meta-training and meta-testing data are drawn from the same distribution. While the authors are well within their rights to confine their study to scenarios where this assumption holds, it would render their setting somewhat artificial and inapplicable to many real-world situations. I believe that assumptions should be dictated by the problem rather than the approach. For meta-learning, it makes sense to assume that meta-training and meta-testing data are IID due to realistic scenarios where this is generally valid, such as few-shot learning problems. However, when it comes to continual learning, it seems unreasonable to assume the same distribution between meta-training and meta-testing data, as non-IID data is precisely what makes CL unique.
> >
> >
> > For these reasons, I am regrettably not convinced that the proposed methods are effective MCL methods in their current form. It remains to be seen whether this method can scale to longer sequences, generalize to OOD data, and compete with other conventional CL approaches.
> >
> >
> > That being said, I respect the majority's opinion, and I am fine with accepting this paper if the AC so decides. However, I will maintain my rating, as the paper does not yet meet my personal standard for a NeurIPS publication.

---

> > > ### Author Response · Authors · 2023-08-20
> > >
> > > We truly appreciate the reviewer for the additional time and effort to respond.
> > >
> > > #### **Our Approach Can Handle OOD**
> > > We think we slightly misinterpreted the reviewer’s concern and provided a partly misleading explanation about OOD in our previous rebuttal. As pointed out by the reviewer, each task in a CL episode is generally OOD and has not appeared previously. This key property of CL also holds in our setting too. Specifically, all tasks that appear during the meta-test are OOD at the task level, i.e., they have never been seen before, even in the meta-training phase. In classification benchmarks, the sets of classes in meta-training and meta-test sets are completely disjoint. In this perspective, our approach can handle OOD data, and we have already demonstrated it in our experiments. Our approach can utilize the strong generalization ability of modern sequence models, which is especially highlighted in natural language domains.
> > >
> > > What we argued in the rebuttal and the paper is about OOD at the episode level, but we think this caused unnecessary confusion. If we construct the episodes of meta-training and meta-test in a similar manner, we can consider their *meta*-distributions of the *episodes* (not their constituent tasks) as the same, even if the individual tasks do not overlap at all. We apologize for the confusion and will improve the explanation.
> > >
> > >
> > > #### **CL vs. MCL**
> > > We fully agree that the ultimate goal of MCL may not be different from CL. However, the existence of the meta-training set in MCL is not just a methodological characteristic but a fundamental difference in problem setting. And this difference draws a clear border between CL methods and MCL methods: CL methods do not have a mechanism to utilize the meta-training set, while its existence is a fundamental assumption of MCL methods. This is why we, along with all the previous works on MCL, refrain from comparing with CL methods.
> > >
> > > #### **Why We Should Take MCL Seriously**
> > > Since the reviewer’s criticism is not limited to our work but extends to all the prior works on MCL, we would like to share our thoughts on why MCL research is important.
> > >
> > > Knoblauch et al. [1] rigorously proved that continual learning, in general, is an NP-hard problem; it is impossible to design a CL algorithm that works universally well in any CL episode. Even in the case of humans, we are naturally good at continually learning some tasks (e.g., memorizing faces) but terrible at others (e.g., memorizing digits).
> > >
> > > In this regard, for a CL algorithm to perform well, there must be some structure in the CL episode, and the CL algorithm should have prior knowledge to exploit it. We think humans’ CL ability has been meta-optimized for the skills that are useful for survival and reproduction (e.g., memorizing faces). To implement such an ability in an artificial agent, there are two choices: (i) manually designing a CL algorithm with a human prior and (ii) designing an MCL algorithm to let it learn the structural prior from meta-training data. The latter better aligns with Sutton’s *The Bitter Lesson*.
> > >
> > > Therefore, we strongly believe that MCL research should continue, despite the current limitations. We hope the reviewer understands that MCL research is still at an early stage, and the limitations (especially the scalability) can be resolved by advances in hardware or sequence modeling technologies.
> > >
> > > [1] Knoblauch et al., Optimal Continual Learning has Perfect Memory and is NP-HARD, ICML 2020.

---

### Official Review · Reviewer_N8DE · 2023-07-10

**Soundness:** 4 excellent
**Presentation:** 4 excellent
**Contribution:** 2 fair
**Rating:** 6
**Confidence:** 3

**Summary:**

This work looks at treating the meta-continual learning problem as instead a sequence modeling problem. Instead of traditional approaches that train a model with an inner loop and then compute a meta-gradient in the outer loop, they replace the inner loop with just inference in a sequence model. The meta gradient step is replaced by a normal gradient step that is taken based on the sequence seen by the model. In order to make this approach work, the paper uses transformers that (1) use causal attention (ie the information transfer only happens forward in time) (2) make use of kernel-based transformers to address the quadratic memory usage of traditional transformers. They test their approach on 4 classification datasets and 3 regression problems.

**Strengths:**

- The results show that there is potential for this approach of treating meta-continual learning as sequence modeling. The performance is generally competitive or superior to prior approaches.

- The presented method is memory efficient and fast compared to previous MCL approaches, at least when dealing with typical MCL benchmarks.

- The paper is well presented and readable, with helpful figures.

**Weaknesses:**

- The paper does mention that their model tends to overfit on the CIFAR-100 dataset. The paper hypothesizes that this is because task diversity is lower, but this is not verified experimentally.
- The novelty is a bit limited, as the paper simply applies transformers to this task of MCL, but still good enough.

**Questions:**

- Do you have results showing how much the performance drops off with efficient transformers compared to the standard one as the data size increases?
- Do you see different performance trends when you increase data per task as opposed to number of tasks?

**Limitations:**

The paper discusses the performance tradeoff that occurs with using efficient transformers to handle long sequences. I think this is a sizable problem, as the method would likely start underperforming or be unscalable when given more data per sequence. They do point out, however, that since they are using standard transformer models, as progress is made with those architectures, that should map well to their approach.

---

> ### Author Rebuttal · Authors · 2023-08-09
>
> We thank the reviewer for the encouraging and insightful comments.
>
> #### **Meta-Overfitting and the Task Diversity**
>
> In the context of classification benchmarks, the term “task diversity” mostly refers to the number of classes. We apologize for not using a clearer description. At the top of Table 1, we highlighted the number of classes in each benchmark to show that the number of classes is strongly correlated with the degree of meta-overfitting.
>
> #### **Novelty**
> We respectfully argue that the main novelty of our work is the reformulation of MCL as a conventional sequence modeling problem. The use of Transformers is just a specific example. Our work opens the possibility that any sequence model that will come in the future can be used as an MCL solver as it is.
>
> ---
> ### Questions
> #### **Efficient Transformers vs. Standard Transformers with More Data**
> Yes, we do have results. In Appendix C, we present a lot more experiments analyzing the various aspects of our approach. Among them are the experiments with longer episodes (Appendix C.2). We test the standard Transformer, Linear Transformer (the best efficient Transformer), and OML (the best baseline). The errors due to longer episodes increase in the order of Transformer < Linear Transformer < OML.
>
> #### **Increasing Examples per Task vs. Increasing the Number of Tasks**
> This is an interesting question. To check the trends when the number of examples per class increases, we additionally tested OML, Transformer, and Linear Transformer in 20-task 25-shot settings. The results are summarized below with 20-task 5-shot and 100-task 5-shot results for comparison. Every score represents an error (the lower, the better).
>
> [CASIA Classification (%)]
> | Method | 20-task 5-shot | 20-task 25-shot | 100-task 5-shot |
> |---|---:|---:|---:|
> | OML | $3.2^{\pm 0.1}$ | $2.4^{\pm 0.0}$ | $6.8^{\pm 0.9}$ |
> | Transformer | $0.4^{\pm 0.0}$ | $0.3^{\pm 0.0}$ | $1.0^{\pm 0.0}$ |
> | Linear TF | $0.7^{\pm 0.0}$ | $0.5^{\pm 0.0} $| $2.3^{\pm 0.1}$ |
>
> [Rotation]
> | Method | 20-task 5-shot | 20-task 25-shot | 100-task 5-shot |
> |---|---:|---:|---:|
> | OML | $0.971^{\pm0.046}$ | $0.994^{\pm0.004}$ | $0.990^{\pm0.008}$ |
> | Transformer | $0.040^{\pm0.001}$ | $0.033^{\pm0.001}$ | $0.031^{\pm0.001}$ |
> | Linear TF | $0.075^{\pm0.002}$ | $0.069^{\pm0.005}$ | $0.047^{\pm0.002}$ |
>
> [Completion]
> | Method | 20-task 5-shot | 20-task 25-shot | 100-task 5-shot |
> |---|---:|---:|---:|
> | OML | $0.1092^{\pm0.0002}$ | $0.1079^{\pm0.0002}$ | $0.1087^{\pm0.0001}$ |
> | Transformer | $0.0999^{\pm0.0002}$ | $0.0973^{\pm0.0002}$ | $0.0989^{\pm0.0001}$ |
> | Linear TF | $0.1039^{\pm0.0003}$ | $0.1013^{\pm0.0002}$ | $0.1084^{\pm0.0001}$ |
>
> We found no significant difference in trends. The order of performance is consistent in all experiments: Transformer > Linear Transformer > OML. In all cases, the 20-task 25-shot setting scores better than other configurations since it has a small number of tasks and a large number of shots.

---

> > ### Comment · Reviewer_N8DE · 2023-08-16
> >
> > I appreciate the response, and am satisfied by the answers. I still recommend an accept, however, I will not be raising my score as I do believe there are limitations to their approach, specifically with respect to the ability to handle long sequences of data.

---

> > > ### Author Response · Authors · 2023-08-17
> > >
> > > We are pleased that the reviewer is satisfied with our response. Additionally, we would like to clarify that handling long sequences is a challenge for *present* Transformers. We emphasize that our main contribution is the reformulation of MCL, such that *generic* sequence models can be directly applied to MCL problems.
> > >
> > > We strongly believe that sequence models will be consistently improved to handle longer sequences, as they have been in recent years; for example, the context length of language models like GPT has been increasing at an incredibly fast pace.
> > >
> > > We kindly request the reviewer to consider the limitations of our formulation and the current Transformers independently.

---

### Decision · Program_Chairs · 2023-09-21

**Decision:**

Accept (poster)

**Comment:**

This paper establishes a connection between continual learning and sequence modeling. It is the first work applying sequence modeling to meta continual learning; the idea is very interesting and shows practical benefit in both model simplicity and empirical performance. In addition, the paper is well written and the authors provide code for reproducibility.

After rebuttal, the authors addressed most of the concerns except the ones on scalability to longer sequences and geralizability to OOD data. Considering continual learning usually involves pretty long sequences which may outpace the capacity of mainstream Transformer-based sequence models, the concerns raised by reviewers are very crucial and keen to the weaknesses which trigger careful treatment. Currently, authors have NOT addressed the concerns sufficiently. By weighing the novel solution to meta continual learning more, this paper could be accepted. But authors must further highlight the limitations of their approach in the long or even endless continual learning settings and in real OOD scenarios. Otherwise, this will mislead future readers as if the paper had solved these two important problems.

Since the merits outweigh the flaws, I recommend the paper be accepted, contingent on the required revision.